# Copper as a Collaborative Partner of Zinc-Induced Neurotoxicity in the Pathogenesis of Vascular Dementia

**DOI:** 10.3390/ijms22147242

**Published:** 2021-07-06

**Authors:** Masahiro Kawahara, Ken-ichiro Tanaka, Midori Kato-Negishi

**Affiliations:** Department of Bio-Analytical Chemistry, Faculty of Pharmacy, Research Institute of Pharmaceutical Sciences, Musashino University, Tokyo 202-8585, Japan; k-tana@musashino-u.ac.jp (K.-i.T.); mnegishi@musashino-u.ac.jp (M.K.-N.)

**Keywords:** zinc, endoplasmic reticulum, MAP kinase, calcium homeostasis, mitochondria, synapse

## Abstract

Copper is an essential trace element and possesses critical roles in various brain functions. A considerable amount of copper accumulates in the synapse and is secreted in neuronal firings in a manner similar to zinc. Synaptic copper and zinc modulate neuronal transmission and contribute to information processing. It has been established that excess zinc secreted during transient global ischemia plays central roles in ischemia-induced neuronal death and the pathogenesis of vascular dementia. We found that a low concentration of copper exacerbates zinc-induced neurotoxicity, and we have demonstrated the involvement of the endoplasmic reticulum (ER) stress pathway, the stress-activated protein kinases/c-Jun amino-terminal kinases (SAPK/JNK) signaling pathway, and copper-induced reactive oxygen species (ROS) production. On the basis of our results and other studies, we discuss the collaborative roles of copper in zinc-induced neurotoxicity in the synapse and the contribution of copper to the pathogenesis of vascular dementia.

## 1. Introduction

Copper (Cu) is the third most abundant trace element in the brain. Cu is essential for most life forms and plays crucial roles in various biological functions, including electron transport, oxygen transport and aerobic respiration [1]. Despite its significance, excess free Cu is toxic because Cu produces reactive oxygen species (ROS) in the redox activity between Cu^2+^ and Cu^+^. Increasing evidence has suggested that the dyshomeostasis of Cu is implicated in the pathogenesis of various neurodegenerative diseases, such as Alzheimer’s disease (AD), prion diseases, Parkinson’s disease (PD), dementia with Lewy bodies (DLB), and amyotrophic lateral sclerosis (ALS) [2,3,4,5].

Here, we focus on the link between Cu and the pathogenesis of a vascular type of senile dementia (VD). VD mainly occurs after stroke or ischemia. During transient global ischemia, the deprivation of glucose and oxygen by the disruption of blood flow and the subsequent neuronal excitation causes a sustainable release of glutamate, which causes neurodegeneration and leads to the pathogenesis of VD [6]. It is widely established that excess zinc (Zn) co-released with glutamate plays central roles in ischemia-induced neurodegeneration and the pathogenesis of VD [7]. Copper ions (Cu^2+^) and zinc ions (Zn^2+^) share similar chemical characteristics and bind the same regions of chelators or metal-binding proteins. Therefore, Cu^2+^ acts competitively with Zn^2+^ in several biological functions. The intake of excess Zn causes a Cu deficiency and vice versa [8]. However, we found that the co-existence of a low concentration of Cu^2+^ markedly exacerbated Zn^2+^-induced neuronal death [9]. We have investigated the molecular pathways implicated in Cu^2+^-enhanced Zn^2+^-induced neurotoxicity and demonstrated the involvement of the endoplasmic reticulum (ER) stress pathway and the stress-activated protein kinases/c-Jun amino-terminal kinases (SAPK/JNK) signaling pathway [9,10]. We also demonstrated that Cu^2+^-induced ROS may be an upstream factor of the ER stress pathway and the SAPK/JNK pathway in the neurodegeneration processes [11,12]. Considering that Cu^2+^ and Zn^2+^ co-exist in the synaptic cleft under ischemic conditions, we hypothesize the involvement of Cu^2+^ as a collaborative partner of Zn^2+^-induced neurotoxicity and subsequent pathogenesis of VD. We also discuss the roles of Cu-binding, amyloidogenic proteins in the synapse in the regulation of Cu homeostasis.

## 2. Copper in the Brain

Cu is an essential trace element and abundantly exists in the liver, kidney and brain [1]. Orally digested Cu is absorbed from the gastrointestinal pathway by divalent cation transporter 1 (DMT1) as Cu^2+^ or by copper transporter 1 (CTR1) as Cu^+^. Cu binds to ceruloplasmin and is then transported in the blood system. The copper-transporting ATPases (ATP7A and ATP7B) play essential roles in Cu distribution in the organs. Cu deficiency or excess due to the impairment of Cu transporters leads to severe neurodegenerative diseases, such as Menkes disease or Wilson’s disease [13]. In the brain, Cu especially exists in the thalamus, substantia nigra, striatum, and hippocampus. Cu plays essential functions in the brain in the synthesis of neurotransmitters, myelination, and neuroprotection against ROS as a cofactor of various enzymes and functional proteins, including cytochrome C oxidase, lysyl oxidase, uricase, dopamine hydroxylase, tyrosinase and Cu/Zn superoxide dismutase (Cu/Zn SOD). Cu is also involved in iron (Fe) homeostasis because ceruloplasmin is a ferroxidase, which converts Fe^2+^ to Fe^3+^.

Increasing evidence suggests that Cu is involved in neurotransmission and information processing in the brain [14,15]. Although the majority of Cu rigidly binds to proteins, a substantial fraction of Cu is in the form of free Cu (Cu^2+^) or loosely bound to small molecular compounds in synaptic vesicles of neurons and is released into the synaptic clefts during neuronal excitation. Secreted Cu^2+^ regulates overall brain excitability by binding to several neurotransmitter receptors, including *N*-methyl-d-aspartate (NMDA)-type glutamate receptors, α-amino-3-hydroxy-5-methylisoxazole-4-propionic acid (AMPA)-type glutamate receptors, γ-aminobutyric acid (GABA) receptors, and purinergic receptors. The accumulation of Cu in the brain reportedly altered GABA-mediated neurotransmission and promoted impulsive behaviors [16]. ATP7A plays a pivotal role in the transport of Cu into synaptic vesicles and, therefore, contributes to axon outgrowth and synaptogenesis [17].

It is well established that a considerable amount of Zn also accumulates in the presynaptic vesicles of glutamatergic neurons as free Zn (Zn^2+^) or in a loosely bound form [18]. Synaptic Zn^2+^ is released with glutamate during neuronal excitation and binds to various neurotransmitter receptors, including NMDA-type glutamate receptors, AMPA receptors, and GABA receptors. These characteristics of Zn^2+^ are quite similar to those of Cu^2+^.

It is possible that—but has yet to be determined whether—secreted Cu^2+^ and Zn^2+^ can diffuse across the synaptic cleft, spill over to neighboring synapses, and modulate the activity of those neighboring synapses, as shown in Figure 1. Synaptic Zn^2+^ plays critical roles in information processing and memory formation [19,20]. It is plausible that differing concentrations of excitatory glutamate and inhibitory Cu^2+^ and/or Zn^2+^ in adjacent synapses transmit spatio-temporal information about neuronal firings and facilitate the precise modulation of neuronal activity. This modulation of neuronal activity at adjacent synapses generates contrasting signals that may enable lateral inhibition and serve as the basis for synaptic plasticity [21]. Therefore, it is possible that “Cu signaling” and “Zn signaling” coordinate at the synapse and modulate the neuronal information [22,23]. Indeed, Cu deficiency caused impaired maturation of the hippocampus in immature rats [24].

Although the concentration of Cu and Zn in the cerebrospinal fluid (CSF) has been reported to be less than 1 μM [25], the concentration in the synaptic cleft may be much higher compared with that in the CSF. The synaptic cleft is a small compartment conceptualized as a cylinder with a radius of 120 nm and a height of 20 nm, and the total volume is estimated to be about 1% of the extracellular space [26]. Thus, it is possible that the concentrations of Cu^2+^ and Zn^2+^ are much higher compared with the CSF. Indeed, the concentration of glutamate in the synaptic cleft is estimated to reach the mM range after 1 ms of neuronal depolarization. Although the exact levels of Zn and Cu in the synaptic cleft remain controversial, the Zn concentrations are estimated to be 1–100 μM [27]. Kardos et al. found ~100 μM of Cu released into synaptic clefts by atomic absorption [28]. However, considering the chemical characteristics of Cu^2+^, it is plausible that synaptic Cu^2+^ loosely bound to small compounds such as organic acids and ATP. A study using a Cu-sensitive fluorescent probe demonstrated that ~3 μM of Cu is released in the synaptic clefts [29].

## 3. Vascular Dementia and Zinc

Senile dementia is a serious problem for our rapidly aging society. It is characterized by profound memory loss and the inability to form new memories in older adults, and its prevalence increases with age. The number of patients in Japan, including those with a mild cognitive deficiency, is estimated to be more than 800 million.

Senile dementia is mainly divided into Alzheimer’s disease (AD), vascular dementia (VD), and dementia with Lewy bodies (DLB). Both AD and DLB are characterized by the deposition of abnormally accumulated proteins: β-amyloid protein (AβP) in AD and α-synuclein (α-Syn) in DLB. VD accounts for approximately one-third of senile dementia cases in Japan. VD is a degenerative cerebrovascular disease that is mainly caused by a series of strokes or ischemia [6]. Risk factors for VD include age, male sex, diabetes, and high blood pressure [30]. After transient global ischemia, the interruption of blood flow and its reperfusion cause the deprivation of oxygen and glucose and the production of ROS. Thereafter, abnormal neuronal excitation occurs in large parts of the brain with the excessive release of glutamate into the synaptic clefts. The excess glutamate causes over-stimulation of its receptors, and the successive entry of large quantities of calcium ions (Ca^2+^) triggers the delayed death of vulnerable neurons, such as pyramidal neurons in the hippocampus, which is an area associated with learning, memory and language. Thus, the development of an infarct and the subsequent cognitive dysfunction characterize the pathogenesis of VD in elderly individuals. An epidemiological study reported the occurrence of dementia symptoms in about 30% of stroke patients within 3 years of the initial stroke [31].

Increasing evidence suggests that Zn has a causative role in neuronal injury after transient global ischemia, ultimately leading to VD [32]. As noted, Zn^2+^ co-accumulates with glutamate in the presynaptic vesicles and is co-released into the synaptic cleft under ischemic conditions. The concentration of Zn^2+^ is estimated to be up to 300 µM [33]. Excess Zn^2+^ reportedly causes the apoptotic death of cultured neurons or neuroblastoma cells (PC12 cells) [34,35]. Koh et al. demonstrated that Zn^2+^ accumulates in apoptotic neurons in the hippocampus after ischemia [36]. “Zn translocation”, namely the entry of Zn from presynaptic vesicles into the postsynaptic neurons, and the increase in intracellular Zn^2+^ levels ([Zn^2+^]_i_), occurs in vulnerable neurons in the CA1 or CA3 regions of the hippocampus prior to the onset of delayed neuronal death following transient global ischemia, enhancing the appearance of the infarct [37]. There are three routes of Zn^2+^ entry into the cell: AMPA-type glutamate receptors, NMDA-type glutamate receptors, and the voltage-gated L-type Ca^2+^ channel (VGLC). Under normal conditions, most hippocampal neurons express AMPA receptors containing the subunit GluR2, which are poorly permeable to Ca^2+^ and Zn^2+^ [38]. However, after ischemia, an acute reduction in the expression of the GluR2 subunit occurs, and neurons express a specific type of AMPA receptor that has channels that are directly Ca^2+^-permeable (Ca-AMPA/kainate channels: Ca-A/K-R). The appearance of Ca-A/K-R channels causes an increased permeability for Ca^2+^ and Zn^2+^ and enhances their toxicity. Therefore, the expression of Zn^2+^-permeable Ca-A/K-R channels and the entry of Ca^2+^ and/or Zn^2+^ through these channels mediate delayed neuronal death after ischemia. Furthermore, the administration of calcium ethylenediaminetetraacetic acid (Ca EDTA), a membrane-impermeable Zn chelator, blocks the translocation of Zn, protects hippocampal neurons after transient global ischemia, and reduces infarct volume [39]. Because Ca EDTA attenuates the ischemia-induced downregulation of the GluR2 gene, Zn is also implicated in the transcriptional regulation of Ca-A/K-R channels. These results strongly suggest that Zn plays a key role in delayed neuronal death after transient global ischemia, a process that is potentially involved in the pathogenesis of VD.

## 4. Copper Enhances Zinc-Induced Neurotoxicity

An understanding of the molecular mechanism underlying Zn^2+^-induced neurodegeneration will advance the development of treatments for VD. We found that Zn causes the apoptotic death of GT1-7 cells (immortalized hypothalamic neurons) in a dose-dependent and time-dependent manner [40]. GT1-7 cells are much more sensitive to Zn and exhibited much lower viability after Zn exposure compared with other neuronal cells, such as PC-12 cells, primary cultured rat hippocampal neurons, and B-50 neuroblastoma cells [41]. The degenerated GT1-7 cells were terminal deoxynucleotidyl transferase-mediated biotinylated UTP nick-end labeling (TUNEL)-positive and exhibited DNA fragmentation. The GT1-7 cells, which were developed by genetical targeted tumorigenesis of mouse hypothalamic neurons, possess neuronal characteristics, such as the extension of neuritis, secretion of gonadotropin-releasing hormone (GnRH), and expression of neuron-specific proteins or receptors, including microtubule-associated protein 2 (MAP2), tau protein, neurofilaments, synaptophysin, GABA receptors, dopamine receptors, and L-type Ca^2+^ channels [42]. Notably, the GT1-7 cells are not subject to glutamate toxicity [8] because they either lack or possess low levels of ionotropic glutamate receptors [43]. Because of these properties, we considered the GT1-7 cell line to be an excellent model system for investigating Zn^2+^-induced neurotoxicity.

First, we examined the effects of treatment with various pharmacological agents prior to Zn^2+^ treatment of GT1-7 cells and found that neither antagonists nor agonists of excitatory neurotransmitters (D-APV, glutamate and CNQX) nor those of inhibitory neurotransmitters (bicuculline, muscimol, baclofen and GABA) influenced the viability of GT1-7 cells [44,45,46,47]. Thus, it is possible that glutamate receptors and/or GABA receptors are not involved in the Zn-induced neurodegenerative pathways. However, several compounds, including energy substrates (pyruvate, citrate), metal chelators (*o*-phenanthroline, deferoxamine), dipeptides (carnosine, anserine), and amino acids (histidine), attenuated the Zn^2+^-induced death of GT1-7 cells.

We examined the effects of other metal ions on Zn-induced neurotoxicity and found that co-exposure with aluminum ions (Al^3+^), gadolinium ions (Gd^3+^), or Ca^2+^ attenuated the Zn^2+^-induced death of GT1-7 cells [13,48]. However, co-existence with Cu^2+^ or nickel ions (Ni^2+^) remarkably exacerbated Zn-induced neurotoxicity [9]. Although cells exposed to Cu^2+^ (CuCl_2_) alone (0–80 μM) did not exhibit neurodegenerative changes after exposure for 24 h, 30 μM of Zn^2+^ caused the loss of 39.0 ± 1.0% of GT1-7 cells (Figure 2). Meanwhile, the co-exposure of 2.5 μM Cu with 30 μM Zn (molar ratio of Cu:Zn = 1:12) resulted in a 76.2 ± 3.3% decrease in cell viability, and 10 μM Cu with 30 μM Zn (molar ratio of Cu:Zn = 1:3) resulted in a 93.8 ± 2.3% decrease in cell viability. Although we applied ZnCl_2_ or CuCl_2_ in the culture media, it is plausible that these metals can bind in a labile manner to small compounds such as citrate or ATP. These results were contrary to our expectation that Cu^2+^ may compete with Zn^2+^-induced neurotoxicity. Therefore, we focused on the molecular mechanism of Cu^2+^-enhanced Zn^2+^-induced neurotoxicity (Cu/Zn neurotoxicity).

## 5. The Molecular Pathways of Copper-Enhanced Zinc-Induced Neurotoxicity

Examination of the molecular pathways involved in the neurotoxicity induced by co-exposure to Cu/Zn may facilitate the development of drugs for the treatment/prevention of VD. Using DNA microarray analysis and real-time PCR (RT-PCR) techniques, we demonstrated that various genes, including metal-related genes (Zn transporter 1 (*ZnT-1*), metallothionein 1 (*MT1*), and metallothionein 2 (*MT2*)), endoplasmic reticulum (ER)-stress-related genes (*CHOP*, *GADD34*), signal-transduction-related genes, and Ca^2+^-signaling-related genes (*Arc*), were upregulated after exposure to Zn^2+^ alone [46,47]. Based on the DNA microarray analysis of GT1-7 cells exposed to Cu^2+^ and Zn^2+^, we found that several genes related to ER stress pathways and MAP kinase pathways were especially upregulated in Cu/Zn conditions compared with Zn alone [9,10,11,12]. Herein, we focus on the five neurodegenerative pathways.

### 5.1. ER Stress Pathway

The ER stress pathway, which impairs ER function and leads to an accumulation of unfolded or misfolded proteins, is implicated in many neurodegenerative diseases, including AD, PD, and cerebral ischemia [49]. ER stress is mediated by three sensors at the ER membrane: PKR-like endoplasmic reticulum eIF2a kinase (PERK), inositol requiring 1 (IRE1), and activating transcription factor 6 (ATF6) [50]. In the PERK branch, activating transcription factor 4 (ATF4) induces C/EBP homologous protein (CHOP), which triggers an intrinsic apoptotic pathway, such as caspase cascades [51], and thereafter CHOP induces GADD34 (protein phosphatase 1 regulatory subunit 15A).

We previously found that Zn^2+^ induces marked upregulation of endoplasmic reticulum (ER)-stress-related genes, especially *CHOP* and *GADD34* in GT1-7 cells [47,48]. We demonstrated that genes related to the PERK branch (*ATF4*, *CHOP* and *GADD34*) were upregulated during Cu/Zn neurotoxicity using RT-PCR analysis [9]. The induction of CHOP protein and the correlation with cell viability were observed by Western blotting analysis. Although Zn^2+^ alone induced these ER-stress-related genes, Cu^2+^ alone did not influence these genes. Therefore, it is possible that the *ATF4*–*CHOP*–*GADD34* axis is responsible for the apoptotic death observed in Cu/Zn neurotoxicity. We found that the ER stress pathway is also involved in Ni^2+^-enhanced Zn^2+^-induced neurotoxicity of GT1-7 cells [52].

### 5.2. SAPK/JNK Pathway

We also found that genes related to the MAP kinase (MAPK) signaling pathway were upregulated after co-exposure to Cu^2+^ and Zn^2+^. The MAPKs are serine/threonine protein kinases that mediate complex signal transduction based on various cellular processes, including proliferation, differentiation, migration, cell death/survival and environmental stress response [53]. There are four subfamilies: the extracellular signal-regulated kinases (ERK1/2), the c-Jun NH_2_-terminal kinases (JNK 1, 2 and 3), the p38 kinases, and the ERK5 (also known as big MAPK-1, BMK1) subfamilies. Among them, the SAPK/JNK signaling pathway plays an important role in apoptotic cell death, necroptosis, and autophagy [54]. The SAPK/JNK signaling pathway is activated by a variety of environmental stressors, such as oxidative stress, inflammatory cytokines and metals. Upon activation of this pathway by various stressors, MAPK kinase 4 (MKK4) or MKK7 phosphorylates and activates SAPK/JNK. Then, c-Jun and activating transcription factor 2 (ATF2), known major downstream factors of SAPK/JNK, are phosphorylated and activated by SAPK/JNK. Ultimately, phosphorylated forms of c-Jun and ATF2 induce downstream factors related to cell death and mitochondrial injury, leading to cell death.

Using RT-PCR and Western blotting, we found that the phosphorylated (i.e., active) forms of SAPK/JNK were increased by co-treatment of Cu^2+^ and Zn^2+^, but not by Zn^2+^ alone nor by Cu^2+^ alone [10]. We found that phospho-c-Jun and phospho-ATF2 were also induced by Cu^2+^ and Zn^2+^. Moreover, an inhibitor of the SAPK/JNK signaling pathway (SP600125) significantly suppressed the induction of CHOP by co-treatment of Cu^2+^ and Zn^2+^ and the activation of the SAPK/JNK signaling pathway and attenuated the neuronal cell death.

### 5.3. Energy Production Pathway

We have already demonstrated that pyruvate, an energy substrate, attenuated Zn^2+^-induced neurotoxicity [40]. Shelline and colleagues reported that Zn exposure decreased the levels of nicotinamide adenine dinucleotide (NAD^+^) and ATP in cultured cortical neurons, and that treatment with pyruvate restored the NAD^+^ level [55,56]. The administration of pyruvate attenuated neuronal death after ischemia in vivo [57]. We found that pyruvate and citrate attenuated Cu/Zn neurotoxicity [58]. Furthermore, the co-existence of pyruvate and citrate did not influence the intracellular concentrations of Zn^2+^ and Cu^2+^ of GT1-7 cells nor the elevations of MTs mRNA. Therefore, it is unlikely that pyruvate and citrate attenuated Cu/Zn neurotoxicity by the chelation to Cu^2+^ and/or Zn^2+^. Therefore, it is possible that the mitochondrial energy production pathway is involved in Cu/Zn neurotoxicity as well as Zn^2+^-induced neurotoxicity.

### 5.4. Disruption of Ca^2+^ Homeostasis

The upstream factors that underlie the Cu/Zn-induced ER stress pathway and the SAPK/JNK pathway are a matter of interest. Here, we focus on two possible upstream pathways: Ca^2+^ homeostasis and ROS production. Both pathways are known to regulate the ER stress pathway and the SAPK/JNK pathway. Notably, disrupted Ca^2+^ homeostasis is also reportedly involved in Zn^2+^-induced neuronal death. Kim et al. reported that Zn neurotoxicity in PC-12 cells was attenuated by an L-type Ca^2+^ channel blocker, nimodipine, and enhanced by the L-type Ca^2+^ channel activator S(-)-Bay K 8644 [35]. Additionally, Zn neurotoxicity was attenuated by aspirin, which prevents Zn^2+^ entry through voltage-gated Ca^2+^ channels (VGLCs) [59].

Using a high-resolution multi-site video imaging system with fura-2 as the cytosolic free calcium reporter fluorescent probe, we previously demonstrated that the exposure of Zn^2+^ caused an elevation in intracellular concentrations of Ca^2+^ ([Ca^2+^]_i_) in GT1-7 cells after 3–30 min of exposure [41]. The addition of Ca^2+^, Al^3+^ and Gd^3+^ attenuated the Zn^2+^-induced death of GT1-7 cells. It is widely known that Gd^3+^ is a blocker of VGLCs [60] and that Al^3+^ inhibits various types of Ca^2+^ channels [61]. We found that pretreatment with Al^3+^ significantly blocked the Zn-induced [Ca^2+^]_i_ elevations and attenuated the Zn^2+^-induced neurotoxicity of GT1-7 cells [41]. These results suggest that Ca^2+^ dyshomeostasis is involved in the mechanism of Zn-induced neurotoxicity. It is plausible that the disruption of Ca^2+^ homeostasis is also involved in Cu/Zn neurotoxicity.

### 5.5. ROS Production

It is widely known that oxidative stress is involved in various neurodegenerative diseases. ROS reportedly induce the ER stress pathway [62], the SAPK/JNK pathway [63], and numerous other adverse effects. As previously noted, an inhibitor of SAPK/JNK (SP600125) suppresses Cu^2+^/Zn^2+^-dependent increases in CHOP expression, suggesting that the ROS–JNK–CHOP pathway is involved in Cu/Zn neurotoxicity. The ROS–JNK–CHOP pathway is also implicated in other types of cell death, such as TNF-related apoptosis-inducing ligand (TRAIL)-induced apoptosis [64].

Cu is a redox-active metal that exists as oxidized Cu^2+^ and reduced Cu^+^, while Zn exists only as Zn^2+^ and is not directly implicated in the redox pathway. The addition of Cu^2+^ induced ROS production in GT1-7 cells, while Zn^2+^ alone did not produce ROS or influence Cu^2+^-induced ROS production in GT1-7 cells [10]. We examined the involvement of oxidative stress in activation of the SAPK/JNK signaling pathway and found that human serum albumin–thioredoxin fusion protein (HSA-Trx), an antioxidative protein, suppressed activation of the SAPK/JNK signaling pathway, inhibited ROS production, and attenuated Cu/Zn neuronal death of GT1-7 cells [11]. Furthermore, selenomethionine (Se-Met), an endogenous selenium (Se)-containing amino acid, induced glutathione peroxidase and blocked ROS production [12]. Pretreatment with Se-Met significantly suppressed the induction of CHOP and attenuated Cu/Zn neurotoxicity.

## 6. Hypothetical Scheme Regarding Cu/Zn Neurotoxicity

On the basis of these findings, we established a hypothetical scheme regarding Cu/Zn neurotoxicity and the implication of Cu and/or Zn in the pathogenesis of vascular dementia (Figure 3).

Under normal conditions, Zn^2+^ and/or Cu^2+^ are secreted into the synaptic cleft during neuronal excitation in a distinctive manner and modulate neuronal information processing. The co-existence of Zn^2+^ and Cu^2+^ in the same synapse does not often occur because the secreted Zn^2+^ and/or Cu^2+^ experience rapid re-uptake by Zn transporters or by CTR1.

However, under pathogenetic conditions such as transient global ischemia, long-term neuronal excitation occurs in major parts of the brain and, thereafter, Zn^2+^ and Cu^2+^ are released and abundantly co-localize in the same synaptic clefts. Excess Zn^2+^ induces a reduction in the expression of the GluR2 subunit, and the elevation in [Zn^2+^]_i_ and [Ca^2+^]_i_ occurs through Ca-A/K-Rs, NMDA-Rs and VGLCs. Al^3+^, a known Ca^2+^ channel blocker, blocks Zn^2+^-induced elevation of [Ca^2+^]_i_ and attenuates Zn^2+^-induced neuronal death. Thereafter, the elevation in [Zn^2+^]_i_ and [Ca^2+^]_i_ triggers the PERK branch of the ER stress pathway, induces CHOP, and finally causes neuronal death. The increase in [Zn^2+^]_i_ also inhibits NAD^+^ in the mitochondrial energy production pathway and causes the depletion of ATP, which leads to neurodegeneration.

When Cu^2+^ and Zn^2+^ co-exist in the same synaptic cleft, Cu^2+^ produces ROS, which subsequently induces the PERK pathway and the SAPK/JNK pathway. Thereafter, both pathways induce CHOP and ultimately exacerbate Zn^2+^-induced neuronal death. SP600125, an inhibitor of the SAPK/JNK pathway, inhibits these processes and attenuates Cu/Zn neurotoxicity. Antioxidants such as HSA-Trx or Se-Met inhibit ROS production by Cu^2+^ and attenuate Cu/Zn neuronal death. In conclusion, synaptic Cu^2+^ collaborates with Zn^2+^ in the same synapse and the co-existence of Cu^2+^ and Zn^2+^ triggers neuronal death after transient global ischemia, ultimately causing the pathogenesis of VD. This hypothesis can explain various aspects of Cu/Zn neurotoxicity and the involvement of synaptic Cu^2+^ and Zn^2+^ in the pathogenesis of VD. Several epidemiological findings suggest that elevated serum Cu is a risk factor of stroke [65,66,67].

Considering the abundance of Cu^2+^ and Zn^2+^ in the synapse, regulatory factors of metal homeostasis may exist in the synapse and play critical roles in the pathogenesis of VD. Several Cu-binding proteins are reportedly localized in the synapse [4,68]. Normal cellular prion protein (PrP^C^) exists at postsynaptic membranes. PrP^C^ binds to Cu at the *N*-terminal octarepeat domain and contributes to intracellular uptake of Cu [4]. PrP^C^ acts as a ZIP Zn transporter analogue and regulates cellular Zn uptake combined with the AMPA receptor [69]. The ZnT-1 Zn transporter is also localized to postsynaptic membranes binding with glutamate receptors [70]. Therefore, both PrP^C^ and ZnT-1 control Zn^2+^ levels at the synapse. PrP^C^ also possesses ferrireductase activity to convert Fe^3+^ and Fe^2+^ and is involved in cellular uptake of Fe^2+^ [71]. Amyloid precursor protein (APP) is localized in the presynaptic membranes and possesses Cu- and/or Zn-binding domains. APP can convert Cu^2+^ to Cu^+^ and contributes to the cellular uptake of Cu [72]. APP is also implicated in Fe^2+^ efflux by binding with ferroportin [73]. α-Syn is a Cu-binding protein present in the presynaptic domain. Cu enhanced the oligomerization of α-Syn [74]. α-Syn also possesses ferrireductase activity to convert Fe^3+^ and Fe^2+^ [75]. Interestingly, all these proteins are involved in the pathogenesis of neurodegenerative diseases [68]. PrP^C^ and its conformational changes are central to the pathogenesis of prion diseases, including scrapie in sheep, bovine spongiform encephalopathy in cattle, and Creutzfeldt–Jakob disease (CJD) in humans. The accumulation of AβP, which is secreted from APP, is observed in AD brain. The oligomerization and the neurotoxicity of AβP are central in AD. The deposition of α-Syn as Lewy bodies has been characterized in the brains of patients with DLB. These disease-related proteins (amyloidogenic proteins) can act to maintain the homeostasis of Cu, Zn and Fe, while these metals can in turn regulate the expressions and functions of the disease-related proteins. The mRNAs of APP, α-Syn, and PrP^C^ possess an iron-responsive element and, therefore, their expressions are regulated by Fe [76]. Cu regulates the processing of APP [77] and the production and clearance of AβP [78]. Considering that these disease-related proteins co-exist in the synaptic cleft, which is a small compartment filled with Cu^2+^ and/or Zn^2+^, it is possible that these proteins can interact with each other in the maintenance of these metals [79,80]. Nibaldo et al. demonstrated that Cu-binding domains of both APP and PrP^C^ prevent neurotoxicity of Cu [81]. AβPs reportedly inhibit the binding of Cu^2+^ to PrP^C^ [82]. PrP^C^ contributes to AD pathogenesis as a toxic receptor of AβP oligomers [83]. Moreover, recent studies have suggested the involvement of PrP^C^ in ischemia-induced neurotoxicity [84,85].

Metal–metal or metal–protein crosstalk is complex and delicate. Therefore, it is plausible that the disorder of these regulatory proteins and disrupting Cu and/or Zn homeostasis may trigger various neurodegenerative diseases, including AD, prion disease, DLB and VD.

There are other regulatory factors of metal homeostasis in the synapse. Metallothionein-3 (MT-3) and carnosine (β-alanyl histidine) are secreted from glial cells or from neurons to the synaptic cleft [86,87]. Carnosine is an endogenous dipeptide that possesses various neuroprotective functions, including anti-oxidant, anti-crosslinking, and anti-glycation functions [8,88]. Carnosine has the ability to chelate to Zn^2+^ and/or Cu^2+^ [88]. The complex of Zn and carnosine (Poraprezinc) is widely used for the treatment of gastric ulcers. We have previously demonstrated that carnosine attenuated Zn^2+^-induced neuronal death [47]. Administration of carnosine was effective in the animal model of ischemic stroke [89]. Dietary supplementation of carnosine and anserine was revealed to improve the cognitive decline and maintenance of memory in elderly people [90]. Based on our findings regarding carnosine, we obtained a patent about carnosine and related compounds as a possible strategy for the prevention and treatment of VD [91,92].

## 7. Conclusions

Our findings regarding the role of Cu in Zn-induced neurotoxicity and ischemia-induced neuronal death clarify the role of Cu as a collaborator with Zn in the pathogenesis of vascular-type dementia. The role of Cu in the synapse may enhance our understanding of the pathogenesis of VD and other neurodegenerative diseases. Substances that attenuate Cu/Zn neurotoxicity may lead to strategies for the prevention or treatment of VD. Further research that provides a more detailed analysis of Cu/Zn neurotoxicity and preventive substances is necessary.

## Figures and Tables

**Figure 1 ijms-22-07242-f001:**
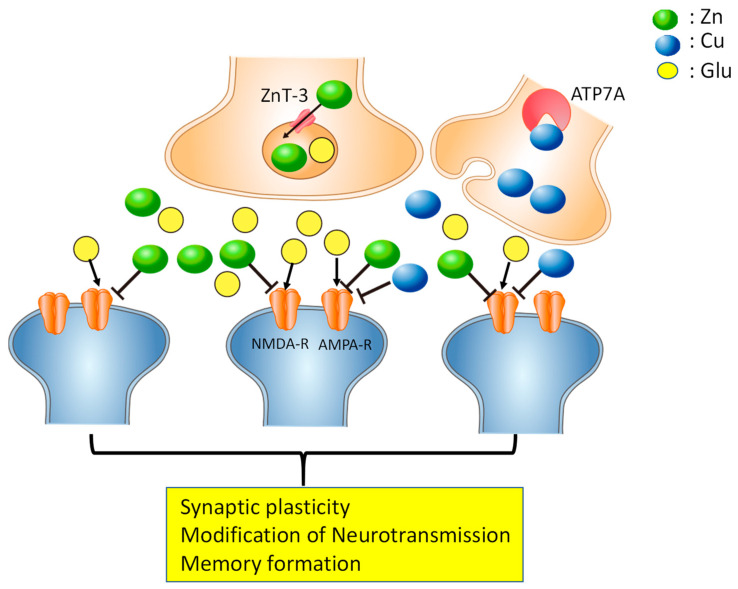
Copper and zinc in the synapse. Under normal conditions, Cu^2+^ and Zn^2+^ are stored in presynaptic vesicles, released with neurotransmitters such as glutamate, and bind to *N*-methyl-d-aspartate (NMDA)-type glutamate receptors (NMDA-R), α-amino-3-hydroxy-5-methylisoxazole-4-propionic acid (AMPA)-type glutamate receptors (AMPA-R), or other receptors. ZnT-3, a Zn transporter, and the copper-transporting ATPase (ATF7A) are involved in the accumulation of Zn and Cu in the synapse, respectively. Cu^2+^ and Zn^2+^ may spill over to the neighboring synapses and modulate excitability, and are implicated in the maintenance of synaptic plasticity and memory formation.

**Figure 2 ijms-22-07242-f002:**
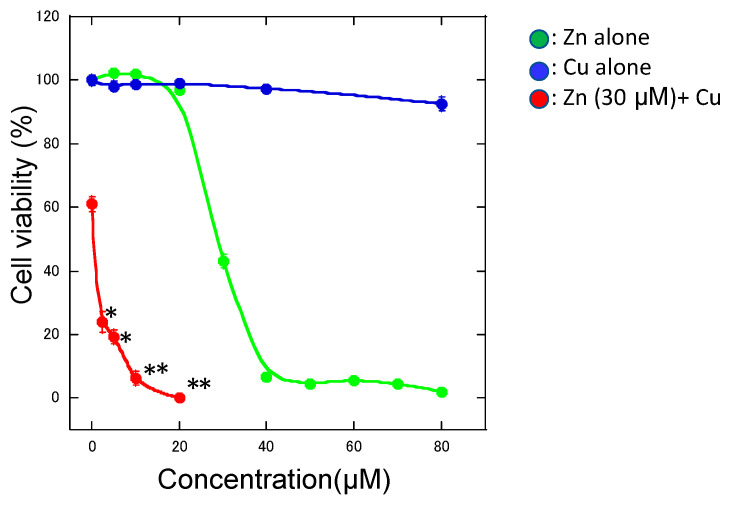
Cu^2+^-enhanced Zn^2+^-induced neurotoxicity. Various concentrations of CuCl_2_ alone (blue circle), ZnCl_2_ alone (green circle), and CuCl_2_ (0–20 μM) with 30 μM ZnCl2 (red circle) were administered to GT1-7 cells in serum-free culture media. After 24 h, cell viability was determined using the CellTiter-Glo^®^ assay. Data are presented as mean ± S.E.M. * *p* < 0.05, ** *p* < 0.01 versus 30 μM ZnCl_2_ alone (Tukey’s test).

**Figure 3 ijms-22-07242-f003:**
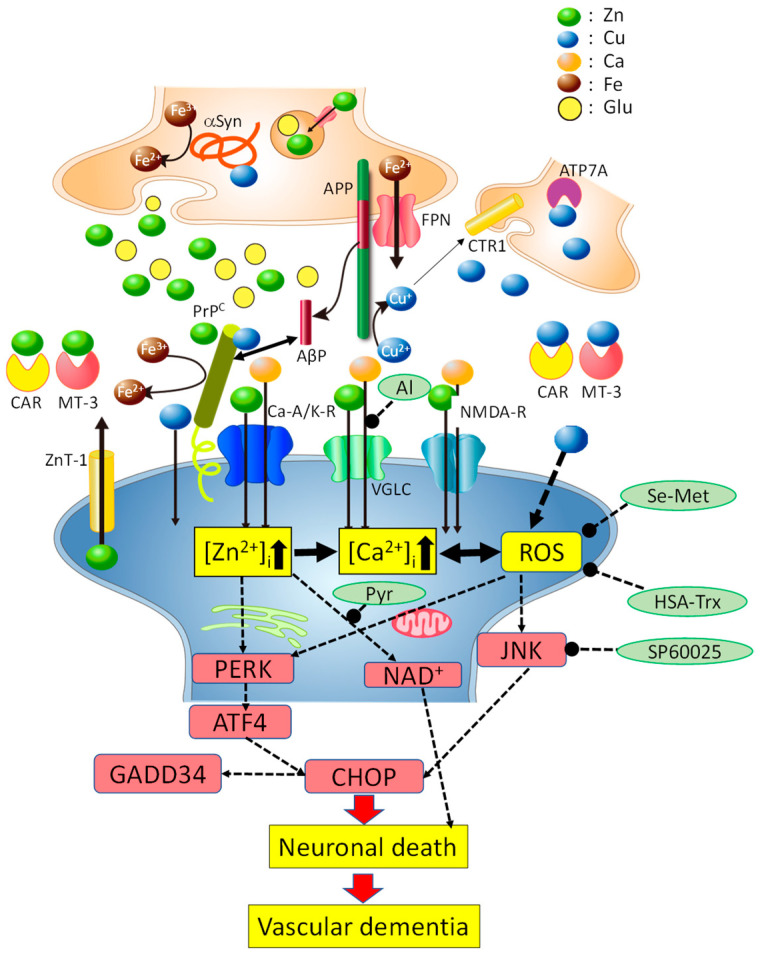
Hypothetical scheme regarding Cu/Zn neurotoxicity. Under pathological conditions such as transient global ischemia, excess Cu^2+^ and Zn^2+^ co-exist in the synaptic cleft. The elevation in [Zn^2+^]_i_ and [Ca^2+^]_i_ triggers ER stress pathways, inhibits the energy production pathway in mitochondria, and induces neurodegeneration. The co-existence of Cu^2+^ with Zn^2+^ causes the production of ROS, upregulates the ER stress pathway and the SAPK/JNK pathway, and finally exacerbates neuronal death. Cu-binding proteins, including normal cellular prion protein (PrP^C^), amyloid precursor protein (APP), and α-synuclein (α-Syn), are located in the synapse and regulate the levels of metals such as Cu, Zn and Fe. Additionally, PrP^C^ regulates Zn^2+^ levels as a ZIP Zn transporter analogue with the ZnT-1 Zn transporter, which is also localized to postsynaptic membranes. PrP^C^ can provide Cu to APP or other Cu-binding proteins in the synapse. APP is mainly localized to the presynaptic membrane. APP binds to Cu and/or Zn and can convert Cu^2+^ to Cu^+^. APP can provide Cu^+^ to CTR1 or other Cu^+^-binding proteins. APP also regulates Fe^2+^ efflux from cells via ferroportin. α-Syn is mainly localized to the presynaptic domain and binds Cu, Mn, and Fe. Both PrP^C^ and α-Syn have ferrireductase activity and provide bioavailable Fe^2+^ to enzymes at the pre- and post-synapse, respectively. Other metal-binding factors such as metallothionein 3 (MT-3) and carnosine (Car) are secreted into synaptic clefts and play critical roles in the maintenance of metal homeostasis. NMDA-R, NMDA-type glutamate receptor; Ca-A/K-R, Ca^2+^-permeable AMPA/kainate-type glutamate receptor; VGLC, voltage-gated L-type Ca^2+^ channel; FPN, ferroportin. The colored circles represent Zn, Cu, Fe, Ca and glutamate.

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
