# Peer review of "Copper as a Collaborative Partner of Zinc-Induced Neurotoxicity in the Pathogenesis of Vascular Dementia"

_ijms, 2021, doi:10.3390/ijms22147242_

Round 1

Reviewer 1 Report

This review is centered on the research of the authors concerning Zn-induced neurotoxicity related specifically to vascular dementia.  It summarizes their work on the effects of Cu added to that of Zn, based on studies with an in vitro cell culture model of immortalized hypothalamic neurons (GT1-7 cells), in which they demonstrated a synergistic enhancement of the Zn-induced neurotoxicity with addition of CuCl2. They put this into the context of what is known about copper and zinc that is related to nerve function and firing, which in itself is something useful to have summarized. Their hypothesis is that Cu, also released during firing of nerves into the synaptic clefts of certain neurons exacerbates the toxic effects of Zn that is released simultaneously, by promoting production of reactive oxygen species (ROS).  They provide many kinds of evidence that support their hypothesis.  At the same time, there are potential limitations in the case of some of this evidence which need to be included and considered, and also a few misinterpretations of published data that must be addressed.  Otherwise the review is generally well written and articulated.

Concerns are that:

  • The approach the investigators/authors have taken to investigate the Zn/Cu effects has certain limitations with regard to what is likely to occur in vivo (vs. in vitro). For example, they are using actual metal ion salts in their studies, and at very high concentrations, to get their effects.  In the case of Zn2+ that is not so much of a problem, but in the case of Cu ions it certainly is.  Cu ions are almost never “free” in the fluids of cells and organisms.  Moreover, in aqueous solution without chelators, they tend to precipitate out as oxides.  In vivo, Cu ions are always bound to carriers, especially proteins, and usually with very high affinity, so that in real life they are very unlikely to promote formation of ROS (through Fenton-type processes.)  The authors themselves are aware of the fact (and describe) that a variety of proteins associated with nerves and synapses bind Cu ions – and some very tightly (like metallothionein, and albumin – which is also present in the brain).  So, concentrations of free Cu ions are very unlikely to approach those achieved in their in vitro studies with cultured cells.
  • In this connection as well, the authors state that the concentrations of Cu ions that have been estimated to be present in synapses after nerve firing is 15 uM, citing Hopt et al. However, those authors actually estimated the concentration as ~3.2 uM, with 15 uM Cu in synaptosomes.  In their own studies on Zn neurotoxicity, the authors of this review found that it took 20 uM Cu2+ to significantly enhance the neurotoxicity of 20 uM Zn2+, although with 30uM Zn it only took only 2.5 uM Cu or more. 
  • Another question is whether the Cu released from synaptosomes is actually in ionic form, and that is a whole other matter that needs to be sorted out.  It seems much more likely that this Cu is also a complex, and whether that complex has the same effect as actual ionic Cu would thus be unclear.  (Just because Hopt et al. detected Cu with a fluorescent chelator doesn’t mean the Cu was present as the free ion.)
  • The authors state repeatedly in various parts of the review that Cu and Zn compete with each other for binding: “Cu2+ acts competitively with Zn2+ ion general” (line 38 for example, and also line 188: “..because it is widely known that Zn and Cu act in a competitive fashion”). That is not generally the case at all in cells and organisms.  Most Cu-dependent/Cu-binding proteins are highly specific for Cu (the exception being metallothionein).  Cu/Zn SOD (SOD1) is another example, where they are both present, but binding to different sites and doing very different things.

Among the minor issues are the following:

  • Line 52 – Cu accumulates in liver, kidney and brain – not really correct (levels are kept very constant (no accumulation), and of course it is present in all cells/tissues.
  • Line 61: cytochrome c oxidase
  • Line 80: It is possible – but as yet unknown – that secreted Cu…..
  • Figure 1 legend, line 5: “…Cu and/or Zn may spill over….”
  • Line 136: Define “Zn translocation”
  • Line 179: the authors should interpret (draw overall conclusions about) the various results they have just presented – what they imply.
  • Line 257: pyruvate and citrate and many other molecules can bind Cu ions, and in general Cu ions bind to proteins of all kinds, and specific amino acids. But the actual molecules that bind Cu ions in vivo are very much more limited, and these are the molecules (mostly proteins) that have a very high affinity for the Cu, which would not include things like pyruvate and citrate.
  • Lines 306-307: Cu would not be taken up by ATP7A (but would be taken up by CTR1 – misspelled)
  • Lines 328-329: Elevations in serum Cu have nothing to do with concentrations of free Cu ions.  Elevations in serum Cu are usually due to increases in the main Cu binding protein of the blood plasma, ceruloplasmin, which is an acute phase reactant, for example, and has a large number of specific Cu-dependent functions.
  • Line 371: Reference is needed for the statement that carnosine chelates Zn and Cu ions (or insert reference 46?)

Author Response

Thank you very much for your kind comments and for checking my manuscript finely.

In particular, your comments about chemistry of Cu are what we are wondering.  Thanks to your detailed comments, we can improve our manuscripts more precisely.  

  • The approach the investigators/authors have taken to investigate the Zn/Cu effects has certain limitations with regard to what is likely to occur in vivo (vs. in vitro). ~~So, concentrations of free Cu ions are very unlikely to approach those achieved in their in vitro studies with cultured cells.

Thank you very much for your suggestion. I agree that the chemical form of Cu (and Zn) are important in the toxicity. I also agree that Cu2+ (or Cu+) not exist in the culture media. I suppose that our used CuCl2 solution can bind to small molecules such as citrate or other organic acids in the culture media, since we used serum-free (protein-free) DMEM in the toxicity experiments. Moreover, it is possible that Cu2+ we applied in the culture media can changed to Cu+, but not yet determined. I added the comments in line 197-199 (page 5) Figure legends (line 205) to make more clear about this problem.

  • In this connection as well, the authors state that the concentrations of Cu ions that have been estimated to be present in synapses after nerve firing is 15 uM, citing Hopt et al. However, those authors actually estimated the concentration as ~3.2 uM, with 15 uM Cu in synaptosomes.  In their own studies on Zn neurotoxicity, the authors of this review found that it took 20 uM Cu2+to significantly enhance the neurotoxicity of 20 uM Zn2+, although with 30uM Zn it only took only 2.5 uM Cu or more. 

Thank you very much for your comments. This is a matter of concern which I myself wondering. Thank you again for the correction of reference results. It is also controversial about Zn level (less than 1uM to 300 uM). Thus, I corrected and changed the line 99-104, and added other study indicating that Cu (1-100 uM) is released after the firing (Ref28).

About our experiments in Fig.2, we found that 2.5uMCu+30 uM Zn (molar ratio: 1:12, similar to the ratio in synapse) caused significant decrease of cell viability, after only 24h. Addition of 2.5-10 uM of Cu significantly decreased the cell viability. We employed 20uMCu+30 uMZn to exhibit more clear results. We added the explanations in line 194-198, and changed Fig.2 to exhibit significance more clear.

  • Another question is whether the Cu released from synaptosomes is actually in ionic form, and that is a whole other matter that needs to be sorted out.  It seems much more likely that this Cu is also a complex, and whether that complex has the same effect as actual ionic Cu would thus be unclear.  (Just because Hopt et al. detected Cu with a fluorescent chelator doesn’t mean the Cu was present as the free ion.)

Thank you very much for your suggestion. I completely agree the points. I have observed that signal of Al3+ disappeared soon after the addition into culture media using NMR. Although I have not tested Cu in the culture media, It is possible that Cu can binds loosely to small molecular compounds such as phosphate, citrate, ATP, etc. However, it is unlikely that Cu precipitated or bound rigidly to proteins since Hopt can detect Cu using fluorescent probe, I think. I added the explanations in line 66-67.

  • The authors state repeatedly in various parts of the review that Cu and Zn compete with each other for binding: “Cu2+acts competitively with Zn2+ ion general” (line 38 for example, and also line 188: “..because it is widely known that Zn and Cu act in a competitive fashion”). That is not generally the case at all in cells and organisms.  Most Cu-dependent/Cu-binding proteins are highly specific for Cu (the exception being metallothionein).  Cu/Zn SOD (SOD1) is another example, where they are both present, but binding to different sites and doing very different things.

Thank you very much for your comments. I changed line 38 to “in several biological functions”. I also deleted line 188.

  • Line 52 – Cu accumulates in liver, kidney and brain – not really correct (levels are kept very constant (no accumulation), and of course it is present in all cells/tissues.

Thank you for your suggestion. I changed to “abundantly exists”. In line 52.

  • Line 61: cytochrome c oxidase
  • Line 80: It is possible – but as yet unknown – that secreted Cu…..
  • Figure 1 legend, line 5: “…Cu and/or Zn may spill over….”

Thank you very much for suggestions. I corrected as suggested.

  • Line 136: Define “Zn translocation”

I added about ZN translocation in line141-142.

  • Line 179: the authors should interpret (draw overall conclusions about) the various results they have just presented – what they imply.

Thank you for your suggestions. I would like to determine some receptor-mediated system is involved in Zn neurotoxicity or not. I added the explanations in line 184-185.

  • Line 257: pyruvate and citrate and many other molecules can bind Cu ions, and in general Cu ions bind to proteins of all kinds, and specific amino acids. But the actual molecules that bind Cu ions in vivo are very much more limited, and these are the molecules (mostly proteins) that have a very high affinity for the Cu, which would not include things like pyruvate and citrate.

Thank you for your comments. We also thought the chelation may occur. Thus, we examined the intracellular concentration of Zn and/Cu are not changed after pyruvate or citrate. Furthermore, the overexporession of metallothionein induced by intracellular Zn did not changed. Thus, we suggest that it is unlikely that pyruvate etc attenuate Cu/Zn neurotoxicity by preventing the intracellular translocation of Cu and/or Zn. I added comments in line 267-270.

  • Lines 306-307: Cu would not be taken up by ATP7A (but would be taken up by CTR1 – misspelled)

Thank you for your correction. I corrected as suggested (Line319-320).

  • Lines 328-329: Elevations in serum Cu have nothing to do with concentrations of free Cu ions.  Elevations in serum Cu are usually due to increases in the main Cu binding protein of the blood plasma, ceruloplasmin, which is an acute phase reactant, for example, and has a large number of specific Cu-dependent functions.

Thank you for your suggestions about our overestimation. I agree the points and am not sure about the elevation of serum Cu can influence brain Cu. Thus, I noted only results (line 340-341).

  • Line 371: Reference is needed for the statement that carnosine chelates Zn and Cu ions (or insert reference 46?)

Thank you for your suggestion. I added reference 88 and 89, and added comments in line 383-384.

Reviewer 2 Report

Dear Auhtors,

I have read this review manuscript by Kawahara and collaborators with great interest and I find it can add quality and be of impact in the field.

The review is organized in sections. And each section includes updated information on the general model of copper-dependent zinc toxicity at the brain level. The authors highlight, consistently with both anatomical and physiological findings, the relevant relation of both copper and zinc, on one hand, and glutamate, on the other, at the synaptic cleft, which explains the various metal-metal and metal-protein interactions variably described in the latest years. Important is also the reference to the low-molecular-weight molecules such as carnosine, that are emerging as valid interactors-cooperators-modulators-etc. of a number of degenerative brain diseases.

In this respect, I would suggest to extend this section given that several recent papers have well established a physiological interplay between some of the actors' metabolisms indicated by the authors (e.g. copper vs. His-based dipeptides and/or zinc vs. His-based dipeptides), and of their gene networks.  

Major points

None

Minor points

Please check for minor typos (e.g. celluroplasmin at page 2 line 54; e.g. 2 in CuCl2 underscript at page 5 line 184)

There is a missing reference (see [] at page 9 line 375) 

Author Response

Thank you very much for your kind comments and suggestions. We believe we can improve our manuscript much better.

Please check for minor typos (e.g. celluroplasmin at page 2 line 54; e.g. 2 in CuCl2 underscript at page 5 line 184)

There is a missing reference (see [] at page 9 line 375) 

Thank you very much for correction of my mistakes. I corrected the manuscript.